# Influence of Recombinant Codon-Optimized Plasmid DNA Encoding VEGF and FGF2 on Co-Induction of Angiogenesis

**DOI:** 10.3390/cells10020432

**Published:** 2021-02-18

**Authors:** Ilnur I. Salafutdinov, Ilnaz M. Gazizov, Dilara K. Gatina, Ruslan I. Mullin, Alexey A. Bogov, Rustem R. Islamov, Andrey P. Kiassov, Ruslan F. Masgutov, Albert A. Rizvanov

**Affiliations:** 1Research Laboratory Omics Technology, Institute of Fundamental Medicine and Biology, Kazan Federal University, 420008 Kazan, Russia; gatina_dilara@mail.ru; 2OpenLab “Gene and Cell Technologies”, Institute of Fundamental Medicine and Biology, Kazan Federal University, 420008 Kazan, Russia; kiassov@mail.ru (A.P.K.); masgut@gmail.com (R.F.M.); 3Department of Human Anatomy, Kazan State Medical University, 420012 Kazan, Russia; ilnazaziz@mail.ru; 4Department of Orthopaedics, Republic Clinical Hospital, 420064 Kazan, Russia; rusdan@mail.ru (R.I.M.); bogov.jr@gmail.com (A.A.B.); 5Department of Medical Biology and Genetics, Kazan State Medical University, 420012 Kazan, Russia; rustem.islamov@gmail.com; 6Morphology and General Pathology Department, Institute of Fundamental Medicine and Biology, Federal University, 420008 Kazan, Russia

**Keywords:** skin wound, regeneration, transfection, hematopoietic stem cell, umbilical cord blood mononuclear cells, dual gene expression cassette plasmids, skin flap, VEGF, FGF2

## Abstract

**Simple Summary:**

Over the past few decades, several methods have been proposed to stimulate skin wound healing. The most promising of these are gene therapy and stem cell therapy. Our present experiments have combined several approaches utilizing human umbilical cord blood mononuclear cells using cell therapy, and direct gene therapy using genetic constructs to accelerate complete healing of skin wounds in rats. Studies have shown that the transplantation of transfected cells stopped proliferative processes in regenerating wounds earlier than the transplantation of untransfected cells. The use of direct gene therapy using the VEGF and FGF2 genes stimulates the revascularization of the rat cutaneous wound.

**Abstract:**

Several methods for the stimulation of skin wound repair have been proposed over the last few decades. The most promising among them are gene and stem cell therapy. Our present experiments combined several approaches via the application of human umbilical cord blood mononuclear cells (hUCB-MC) that were transfected with pBud-*VEGF*165-*FGF*2 plasmid (gene-cell therapy) and direct gene therapy using pBud-*VEGF*165-*FGF*2 plasmid to enhance healing of full thickness skin wounds in rats. The dual expression cassette plasmid pBud-*VEGF*165-*FGF*2 encodes both VEGF and FGF2 therapeutic genes, expressing pro-angiogenic growth factors. Our results showed that, with two weeks post-transplantation, some transplanted cells still retained expression of the stem cell and hematopoietic markers C-kit and CD34. Other transplanted cells were found among keratinocytes, hair follicle cells, endothelial cells, and in the derma. PCNA expression studies revealed that transplantation of transfected cells terminated proliferative processes in regenerating wounds earlier than transplantation of untransfected cells. In the direct gene therapy group, four days post-operatively, the processes of flap revascularization, while using Easy LDI Microcirculation Camera, was higher than in control wounded skin. We concluded that hUCB-MC can be used for the treatment of skin wounds and transfection these cells with VEGF and FGF2 genes enhances their regenerative abilities. We also concluded that the application of pBud-VEGF165-FGF2 plasmids is efficient for the direct gene therapy of skin wounds by stimulation of wound revascularization.

## 1. Introduction

The regeneration of skin wounds remains a major problem for experimental studies, the relevance of which is due to a high number of patients with acute and chronic mechanical, thermal, chemical, autoimmune damage of skin, and many other related conditions. In many articles and reviews, it has been shown that the regeneration of the epidermis in mammals and humans is always based on the proliferation of viable epithelial cells of skin derivatives, primarily from hair follicles. In the process of regeneration, the elements of the external root sheath and sweat glands form multilayer cords, which gradually shift to the surface of the skin and become superficial epithelium [1,2]. Further studies developed the stem cells theory and identified them in the basal layer of the human epidermis and in hair follicles. In animals, stem cells have been found in the basal layer of the epidermis and tubercles of external root sheath, which are located at the base of infundibulum below the outlet of excretory ducts of sebaceous glands [3]. Modern understanding of skin regeneration, in addition to stem cell proliferation, includes an understanding of cell migration, extracellular matrix accumulation, angiogenesis, and subsequent tissue remodeling [4,5].

Particularly severe forms of skin lesions, or genetic diseases, such as congenital epidermolysis bullosa, require the search for alternative approaches for the treatment of wounds [6,7]. Today, alternative cellular sources of skin tissue regeneration and engineering are being actively studied: bone marrow stem cells [8], cord blood cells [9], adipose tissue [10], skin epithelium and hair follicles [11,12], and induced pluripotent stem cells (iPS cells) [13].

Research is also actively conducted on the introduction of tissue engineering methods, transplantation into skin wounds cells that were transfected with genes of cytokines and growth factors, which are involved in the process of regeneration [14,15,16,17,18,19].

The aim of our work was to study the transplantation of hematopoietic stem cells from umbilical cord blood, simultaneously genetically modified with *VEGF* and *FGF2* genes, as well as direct gene therapy with the dual gene expression cassette plasmid pBud-*VEGF165-FGF2*, during the regeneration of tissues following skin wounds in rats.

## 2. Material and Methods

### 2.1. Umbilical Cord Blood Mononuclear Cells

Cord blood was collected by trained staff at the maternity hospital, according to the instructions of the Stem Cell Bank of Kazan State Medical University (license №ΦC-16-01-001421 from 13.04.2016), after the patients had signed informed consent. The mononuclear fraction of human cord blood was processed by centrifugation in a Ficoll density gradient according to a method described earlier [20]. Counting the number of nucleated cells and the determination of their viability was carried out following staining with trypan blue. In all obtained samples (*n* = 4), the number of viable cells was higher than 97%.

### 2.2. Plasmids Vectors

The dual gene expression cassette plasmid vector pBud-VEGF165-FGF2 was used for direct gene therapy and the transfection of umbilical cord mononuclear cell (UCB-M**C**) by electroporation, as previously described by Rizvanov et al., 2011 [21]. Untransfected UCB-M**C**s served as controls. After transfection, the cells were incubated for 24 h in complete RPMI media (PanEko, Moscow, Russia), supplemented with 10% FBS (HyClone, Logan, UT, USA), L-glutamine (Thermo Fisher Scientific, Waltham MA, USA), and 1% penicillin-streptomycin solution (Invitrogen Life Technologies, Carlsbad, CA, USA). in 95% air (5% CO2) at 37 °C. Real-time quantitative PCR (qPCR) was performed to investigate the expression of the cloned genes.

### 2.3. Quantitative Analysis of Transgenes Expression in Modified Umbilical Cord Blood Mononuclear Cells

The quantitative analysis of VEGF and FGF2 genes in UCB-MC that was transfected with pBud-VEGF165-FGF2 was estimated using real-time polymerase chain reaction (RT-PCR) 72 h after cell modification. The total RNA was extracted with the RNeasy Mini Kit (Qiagen, Hilden, Germany), according to the manufacturer’s instructions. The quality and quantity of isolated RNA were estimated using a Nanodrop ND-2000c (Thermo Scientific, Waltham, MA, USA). Reverse transcription was performed using the six nucleotide random primers and RevertAid Reverse Transcriptase (Thermo Fisher Scientific). The expression of the genes VEGF and FGF2 was analyzed using primers and TaqMan probes (VEGF forward: TACCTCCACCATGCCAAGTG, reverse: TGATTCTGCCCTCCTCCTTCT, TM-prob: TCCCAGGCTGCACCCATGG; FGF2 forward: CCGACGGCCGAGTTGAC, FGF2 reverse: TCTTCTGCTTGAAGTTGTAGCTTGA, TM-prob: CCGGGAGAAGAGCGACCCTCAC). The polymerase chain reaction was performed using a CFX96 thermal cycler (BioRad, Hercules, CA, USA). The level expression of housekeeping gene β-actin (forward: GCGAGAAGATGACCCAGGATC, reverse: CCAGTGGTACGGCCAGAGG, TM-prob: CCAGCCATGTACGTTGCTATCCAGGC) was used for reference. Normalized β-actin expression was calculated with the ΔΔCt (Livak) method for relative quantification [22].

### 2.4. Animals

The study was performed on male Wistar rats weighing 225–250 g (*n* = 20). The animals were kept under standard vivarium conditions under a day/night mode 12/12 with free access to feed and water. All of the experimental procedures were in accordance with the ethical rules, as accepted by Kazan Federal University, and approved by the Local Ethical Committee (Protocol №15, 28.03.2019) and the international bioethical standards that were defined by the International guiding principles for biomedical research involving animals (2012), the directive 2010/63/EC, and the 3Rs principles. 

The animals were divided into four groups: group 1-a rounded skin wound with a diameter of 1 cm and transplantation of 2 × 10^8^ human umbilical cord cells that were transfected with the *VEGF* and *FGF2* genes, group 2-a rounded skin wound with a diameter of 1 cm and transplantation of 2 × 10^8^ untransfected human umbilical cord cells, group 3-skin flap 1 × 9 cm and injection of 500 micrograms of dual gene expression cassette plasmid vector pBud-*VEGF165-FGF2*, and group 4-skin flap 1 × 9 cm and injection of 0.9% NaCl. Figure 1 presents the research strategy.

### 2.5. Modelling and Treatment of Full-Layer Thickness Skin Wounds

All of the surgical procedures were performed under anaesthesia with a 6.4% solution of chloral hydrate (AppliChem GmbH, Darmstadt, Germany) at the rate of 400 mg of dry matter per 1 kg of animal weight. The dorsal hair was shaved, and the residual hair was removed with a depilatory cream. The skin was treated with 70% ethanol solution.

#### 2.5.1. Circular Excision Skin Wound Model

An experiment of the formation of full-thickness skin wounds with a diameter of 1 cm in groups 1 and 2 was performed under anesthesia on the back of the rats with sterile surgical scissors. The day of the injury was considered to be day-zero of the experiment. One day later, the wound perimeter was injected with genetically modified or native human umbilical cord blood cells.

#### 2.5.2. Skin-Fascial Flap Model

The skin flap was clipped at the caudal end, thereby creating the conditions of subcompensated blood supply; thereafter, the flap was stitched back into it’s appropriate anatomical place. Further, along the entire length of the flap either 500 μg of pBud-VEGF165-FGF2 plasmid (600 μL, 12 injections of 50 μL) (group 3), or 0.9% NaCl in the same amounts (group 4) were injected. In group 3 and 4 animals, a 1 cm wide by 9 cm long skin flaps were made along the vertebral column, whilst the animals were anesthetized (Figure 2).

### 2.6. Histological Investigation 

Animals of groups 1 and 2 were terminated from the experiment after two weeks; the sample of regenerated skin wound was excised and put fixed in 10% neutral formalin on 0.2 M phosphate buffer (pH = 7.4) for 24 h, then embedded in paraffin according to standard procedures [23]. 

Tissue sections were studied using immunohistochemically with commercially available antibodies to HLA-ABC (clone W6/32, 1:100, Dako, Copenhagen, Denmark), HNA (clone 235-1, 1:100, Millipore, Billerica, MA, USA), C-kit (clone T595, 1:400, Novocastra, Newcastle, UK), PCNA (clone PC10, 1:100, Dako, Glostrup, Denmark), CD 31 (clone 1A10, 1:20, Leica Biosystems, Newcastle, UK), CD144 (clone 36B5, 1:100, Novocastra, Newcastle, UK), WF (clone NCH-38, 1:50, Dako, Glostrup, Denmark), CD 34 (clone QBEnd/10, 1:75, Novocastra, Newcastle, UK). Immunohistochemical staining was performed with Novolink polymer detection imaging kit by Novocastra (Novocastra^®^, New Castle Upon Tyne, UK) following the manufacturer´s instructions. The control of the specificity of the immunohistochemical reaction was carried out with a non-immune mouse or rabbit serum, as well as by excluding the primary antibodies from the reaction altogether.

#### Morphometric Examination

A morphometric analysis of the content of proliferating keratinocytes was performed while using a morphometric ocular grid at 400× total magnification on a Leica DM 1000 microscope (Leica Microsystems, Wetzlar, Germany).

### 2.7. Laser Doppler Imaging (LDI)

The therapeutic effects of the plasmids were evaluated four days after the operation using Laser Doppler imaging (LDI) EasyLDI Microcirculation Camera, which provided the visualization of perfusion in the dermal microcirculation in real time and enabled both intraoperative assessment and postoperative monitoring of skin flap perfusion (Aïmago SA, Lausanne, Switzerland). There is no wellness hazard that is associated with exposure to the EasyLDI [24].

### 2.8. Statistical Analyses

All of the data are expressed as mean ± standard deviation (SD). Differences between the analyzed groups were assessed by one-way ANOVA. The *p* value of <0.05 is considered to be statistically significant.

## 3. Results and Discussion

### 3.1. Molecular Analysis of Gene Modified UCB-MC In Vitro

Gene expression of VEGF and FGF2 on the mRNA levels that were modified UCB-MC were evaluated while using qPCR. The efficiency of UCB-MC transfection with pBud-VEGF-FGF2 was confirmed 72 h after electroporation. Gene expression analysis revealed a significant change on the mRNA levels of the VEGF and FGF2 genes expression in modified UCB-MC by 57,000 ± 2250 and 19,000 ± 1186 times, respectively, when compared to unmodified UCB-MC (*p* < 0.05) (Figure 3).

### 3.2. Effect of Transplanted Genetically Modified Cells on the Regeneration of Skin Defects Full-Thickness Skin Wounds

Immunohistochemical studies showed that HLA-ABC was determined among the cells of the reticular layer of the dermis of groups 1 and 2 (Figure 4A). 

The expression pattern in all cells was cytoplasmic, and the expression level ranged from low to medium. The study of HNA, the second marker of human cells that was used in our study, revealed HNA-positive cells among cells of epidermis and hair follicles (Figure 4B). Thus, it can be argued that cord blood hematopoietic stem cells are able to participate in the regeneration of skin wounds in vivo. These in vivo data were largely consistent with the in vitro data, which dispensed the ability of umbilical cord blood cells to differentiate into keratinocytes [25].

The study of C-kit expression revealed the presence of many C-kit-positive cells in the dermis in groups 1 and 2. The morphology of cells resembled lymphocytes, with large nuclei and narrow rim of immunopositively cytoplasm (Figure 4C). It can be assumed that human cord blood mononuclear cells after two weeks of transplantation retain the expression of the stem cell marker–C-kit. This assumption is confirmed by the study of the marker of hematopoietic stem cells–CD34. We found the presence of CD34-positive cells similar in morphology to C-kit-positive cells in the dermis of regenerating wounds (Figure 4 D). In addition, CD34 is also a marker of endothelial cells, and we found that some endothelial cells were CD34-positive (Figure 4E). The immune reaction of CD34 antibodies in only some endothelial cells observed can be explained by specific immunoreactity against human CD34 molecules, which is stated in the manufacturer’s specifications, and no cross-reactivity against the rat CD34 molecule. In this regard, it cannot be excluded that transplanted cord blood mononuclear cells under conditions of our experiment are capable of differentiation in the endothelial direction in vivo. In previous studies, we had already identified the angiogenic potential of umbilical cord blood cells in an amyotrophic lateral sclerosis model [18]. The study of other endothelial markers, such as WF (Figure 4F), CD31 (Figure 4G), and CD144 (Figure 4H), also revealed the presence of human antigen immunopositive capillaries. It should also be noted that, whilst studying CD144, we revealed its expression not only in endotheliocytes, but also in hair follicle cells, fibroblasts, and in some mononuclear cells.

PCNA studies revealed proliferation among keratinocytes of the epidermis and hair follicles, dermal cells, and endotheliocytes (Figure 4I,J). A morphometric study of proliferating keratinocytes showed, that in experimental group 1, PCNA expression was significantly lower (19.3 ± 1.6%; *p* < 0.05) than in group 2 (28.7 ± 2.1%; *p* < 0.05). These results indicate that, in group 1 animals, after two weeks of transplantation of hUCB-MC transfected with *VEGF* and *FGF2* genes along the periphery of full-thickness skin wounds, keratinocyte proliferation reached physiological levels of 20.8 ± 1.6%, and in group 2 levels remained high, which indicates incomplete reparative regeneration. Thus, the transfection of cells with the *VEGF* and *FGF2* genes before transplantation accelerated tissue regeneration in skin wounds.

### 3.3. Effect of VEGF and FGF2-Expressing Plasmids on the Blood Flow in the Skin-Fascial Flap 

In the third group, following the operation and the introduction of pBud-VEGF165-FGF2 at an earlier date, as compared with the control (fourth group), there were signs of the cleansing of the formed area of the wound surface, granulation, epithelialization, and a decrease in local infiltration. Heartbeat and blood pressure were retained at a constant level throughout the study in all animals. No changes in temperature and inflammation were observed in any of the long skin flaps. Within four days, despite visual signs of venous insufficiency, we found compensation of the blood supply of 55 ± 8.7% relative to the intact skin in the distal part of the flap, whereas, in group 4, decompensation of microcirculation remained, amounting to 10 ± 5.1% relative to the intact skin (Figure 5A,B).

## 4. Discussion

The management and stimulation of skin regeneration are an actual task of modern medicine because there are constantly many patients with skin wounds of different genesis. The treatment of skin wounds in clinics uses different approaches, which might give better results with quicker vascularization. New methods for stimulation of angiogenesis with growth factors and cytokines, such as VEGF [26,27], FGF [28], hepatocyte growth factor (HGF) [29], platelet-derived growth factor (PDGF) [30], IL-10 [31], were proposed for that purpose, which have already showed promising results. It is still unclear which way of administration of growth factors (proteins, naked genes, viral or non-viral methods of gene delivery, cell-mediated gene delivery) is more efficient for treating full-layer thickness skin wounds.

Recombinant protein therapy might be the most practical means to administer growth factors in the clinic when the doctor knows precisely the dosage and the molecule in the syringe. However, the half-life of recombinant growth factors, conditions of storage, transportation, dosage, and frequency of administration make this treatment method problematic at best. To achieve better outcomes, it might be more beneficial to administer genes coding the protein of interest to achieve better therapeutic results. It can potentially result in sustained local protein secretion with minimal adverse effects. Our previous studies [32,33] have shown that it is usually more beneficial to use cell-mediated gene therapy in transfected administration with several genes hUCB-MC.

In the present study, we used two models of skin regeneration on rats. In one model, we made a rounded skin wound with a diameter of one cm and injected 2 × 108 human umbilical cord cells that were transfected with the VEGF and FGF2 genes (group 1) or the same number of untransfected cells (group 2). One cm in diameter wound is very small in relation to the rat’s whole skin surface area. It gives an excellent opportunity to assess histological changes during skin regeneration in a short period of time. Analyzing the results of immunohistochemical studies, we found that transplanted naïve (un-transfected) hUCB-MC participate in skin regeneration in rats. Comparable results were shown in El-Mesallamy et al. treatment with UCB-MC resulted in cutaneous wound contraction [4].

However, the transplantation of hUCB-MC transfected with the VEGF and FGF2 genes showed better regeneration results of full-layer thickness skin wounds in rats than after the transplantation of naïve hUCB-MC. Ikeda et al., in their study of chronic hindlimb ischemia, showed the same pattern of results. The authors showed that blood flow was significantly improved in nude rats that received untransfected cord blood mononuclear cells. The transplantation of cord blood mononuclear cells transfected with the hVEGF gene yielded greater improvements in blood flow [34]. Cell-mediated gene therapy with using of cord blood cells can also be used in the stimulation of angiogenesis in other pathologies, for example, in myocardial infarction model. Recently, Das et al. reported a promising method of regenerative therapy using transiently overexpressed angiogenic genes VEGF and PDGF in ex vivo expanded human umbilical cord blood-derived CD133^+^/CD34^+^ progenitor cells while using a bicistronic vector, in which both VEGF-A164 and PDGF-BB genes under CMV promoter, to augment stem cell effect for treatment of myocardial ischemia induced in the rat model [35]. We have recently shown that the transplantation of UCB-MC modified lactoferrin promotes maxillofacial area phlegmon recovery and cervical lymph node remodeling in rats [36]. Thus, our current findings provide the feasibility and effectiveness of overexpressing angiogenic growth factors on hUCB-MC for the treatment of skin wounds. Treatment with hUCB-MC transfected with the pBud-VEGF165-FGF2 construct has synergistic effects of hUCB-MC and recombinant growth factors on the recovery of skin wounds in rats. This finding was in harmony with the results of Spanholtz et al., who showed an improvement of skin flap survival by using transfected with VEGF165 and bFGF genes fibroblasts, which resulted in an increase of the number of smooth muscle alpha-actin^+^/CD31^+^ blood vessels and a reduction of necrosis by 25% [37].

In the second model, we made skin flap 1 × 9 cm on the back of the rat with a relatively high length to width ratio (9:1). This is a modified classic rat skin flap model described by McFarlane et al. in 1965 and used to study skin necrosis and its prevention. The original dimensions that were described in McFarlane’s et al. study were 4 cm across the base by 10 cm in length (1:2.5 ratio) [38]. Several studies have cited this method as a reference, but with different parameters. Skin-fascial flaps measuring 1.5 × 7.5 cm [26], 2 × 9 cm [39], 3 × 7 cm [40], and 3 × 9 cm [41], 3 × 10 cm [42] have been reported and attributed to McFarlane casting doubt on the results of the original description. Therefore, it can be argued that our model is adequate, and it can be used for testing of wound healing.

Combined stimulation with VEGF and FGF-2 was proposed as a potent strategy for therapeutic angiogenesis [43]. In our experimental design, we tried to save the skin flap by stimulation of vascularization by the injection of 500 micrograms of dual gene expression cassette plasmid vector pBud-VEGF165-FGF2 (group 3) or 0.9% NaCl (group 4, control). By using laser doppler imaging, we proved that direct gene therapy of skin-fascial flap with dual gene expression cassette plasmid vector pBud-VEGF165-FGF2 improves the vascularization of the skin flap. These results also correspond with our previous findings, which showed high efficiency of treatment of critical lower limb ischemia in patients with direct gene therapy using VEGF and FGF2 genes [44]. Therefore, the simultaneous overexpression of VEGF165 and FGF2 increases the revascularization of the ischemic skin flap, presumably due to the activation of the endothelium’s proliferation [45]. The Bud-VEGF 165-FGF2 group’s blood flow was more efficiently restored and it did not undergo spontaneous involution during the observation. Hence, skin flaps were protected from irreversible ischemic injury.

Notwithstanding the safety of using plasmid vectors and the low cost of their production, several limiting factors were described for the widespread use of plasmid vectors during stimulating angiogenesis. First of all, this has low efficiency in the modification of target cells. Concurrently, various strategies have been developed to increase the efficiency of genetic modification in using plasmid vectors [27,46]. Consequently, further clarification studies are required for more efficient and safe delivery of plasmid systems. Taken together, our results indicate the applicability of modern methods of gene and gene-cell therapy to stimulate the recovery of various skin defects and it can be integrated into clinical practice to treat jeopardized tissues.

## 5. Conclusions

Finding transplanted hUCB-MC and their progeny at the site of regenerating full-thickness skin wounds, based on immunohistochemical studies, concludes that hUCB-MC can be used for the treatment of skin wounds. After the transplantation of hUCB-MC transfected with the VEGF and FGF2 genes, we observed quicker regeneration of full-thickness skin wounds in comparison with transplantation of untransfected hUCB-MC. These data conclude that genetic modification of hUCB-MC with pBud-VEGF165-FGF2 plasmid is efficient and enhances the regeneration of skin wounds. The application of recombinant plasmid construct pBud-VEGF165-FGF2 alone in the site of skin regeneration induces angiogenesis. It has a pronounced stimulating effect on the vascularization, as we have demonstrated in the of decompensated skin-fascial flap model. According to the results of the experiment, it can be argued that the dual gene expression cassette plasmid vector pBud-VEGF165-FGF2 has a pronounced stimulating effect on skin regeneration in direct gene therapy and after the transfection of hUCB-MC, which opens new perspectives in the treatment of nonhealing wounds of various genesis.

## Figures and Tables

**Figure 1 cells-10-00432-f001:**
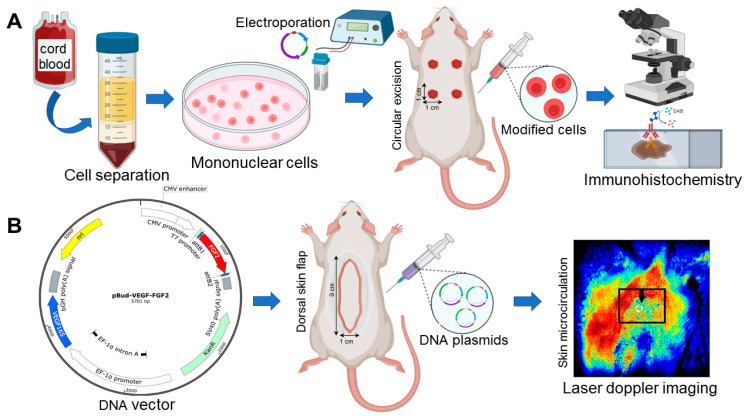
Strategy for investigation. (**A**). Schematic depicting the umbilical cord blood isolation procedure, gene modification and injection cells into the excision skin wound. Cord blood was harvested, separated, washed, transfected recombinant plasmids and transplanted. Skin wounds were examined by immunochemistry. (**B**). Schematic depicting the injection of naked DNA plasmid vectors into the formed dorsal skin flap. Picture of the dorsal side of the 1 × 9 cm skin-fascial flap. Injections DNA plasmids were achieved by injecting 600 μL of 50 μL in 12 evenly dispersed sites skin flap. Skin flap vascularization was examined by laser doppler blood flow imaging. Figure created with BioRender. (https://biorender.com (accessed on 20 December 2020)).

**Figure 2 cells-10-00432-f002:**
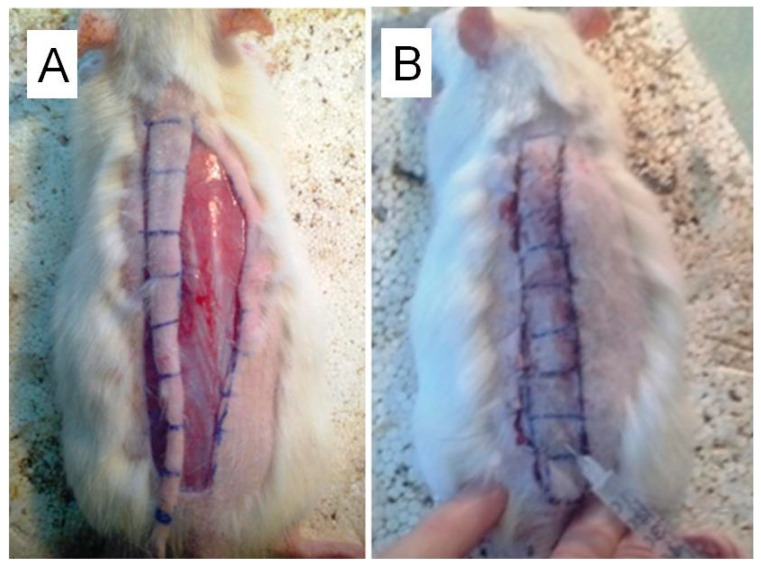
Stages of formation of a dorsal skin flap and the injection of plasmid DNA: (**A**). longitudinal skin flap with cranial supply leg; (**B**). injection of 500 μg of dual gene expression cassette plasmid pBud-*VEGF165-FGF2* in 500 μL saline. skin flap, with 1 × 9 cm dimensions.

**Figure 3 cells-10-00432-f003:**
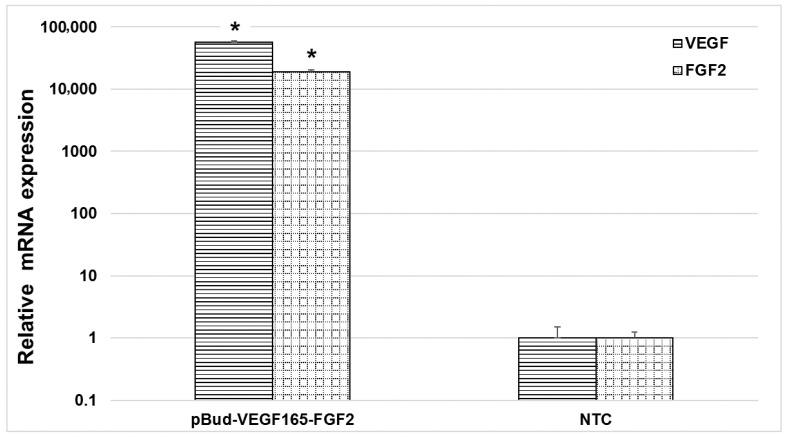
The expression of mRNA of the VEGF and FGF2 genes in human umbilical cord blood mononuclear cells transfected with the Pud-VEGF165-FGF2 plasmids. The relative level of mRNA expression was determined using real-time PCR with specific primers and TaqMan probes. Control-the level of mRNA expression of the examined genes in untransfected cells. The amount of mRNA was normalized to β-actin. Data represent at least two independent experiments and they are displayed as mean ± standard deviation * *p*-value < 0.05.

**Figure 4 cells-10-00432-f004:**
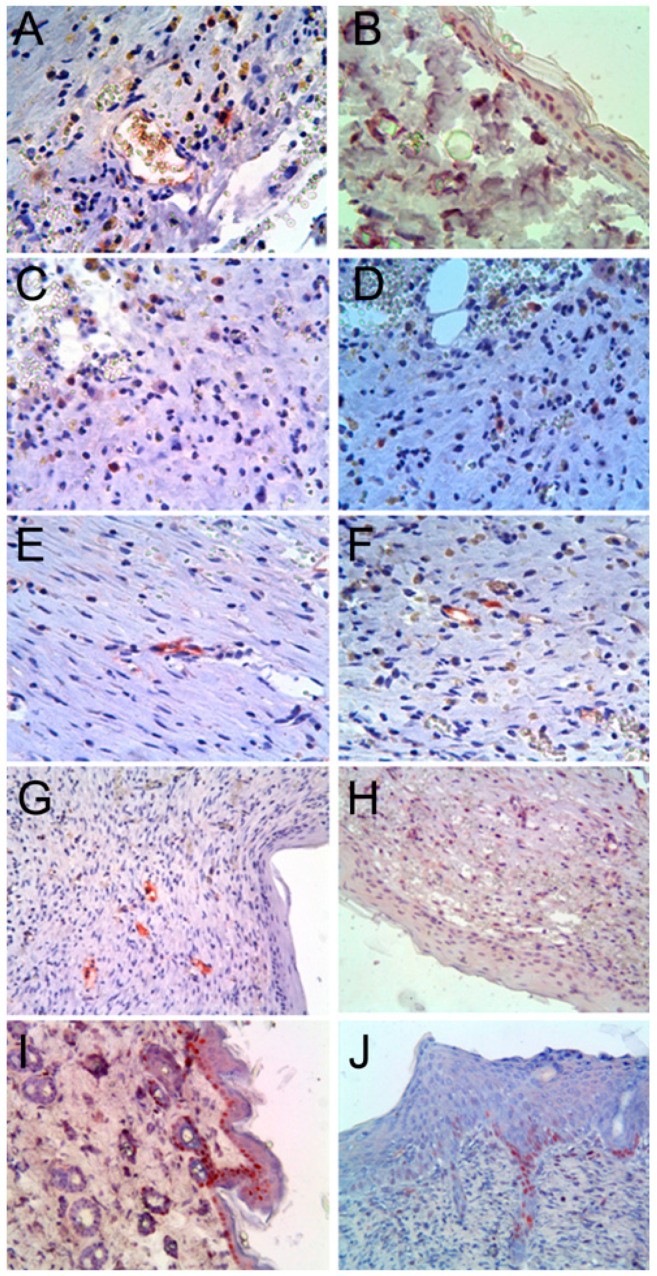
Skin wounds of rats two weeks after injury and treatment; immunohistochemical reactions with antibodies against: (**A**)-HLA-ABC, group 1; (**B**)-HNA, group 2; (**C**)-C-kit, group 1; (**D**)-CD34, group 1; (**E**)-CD34, group 2; (**F**)-WF, group 1; (**G**)-CD31, group 2; (**H**)-CD144, group 1; (**I**)-PCNA, group 2; (**J**)-PCNA, group 1. Nuclei were counterstained with hematoxylin. Group 1-a rounded skin wound with a diameter of 1 cm and transplantation of 2 × 10^8^ human umbilical cord cells transfected with the *VEGF* and *FGF2* genes. Group 2-a rounded skin wound with a diameter of 1 cm and transplantation of 2 × 10^8^ untransfected human umbilical cord cells. Original magnification 400×.

**Figure 5 cells-10-00432-f005:**
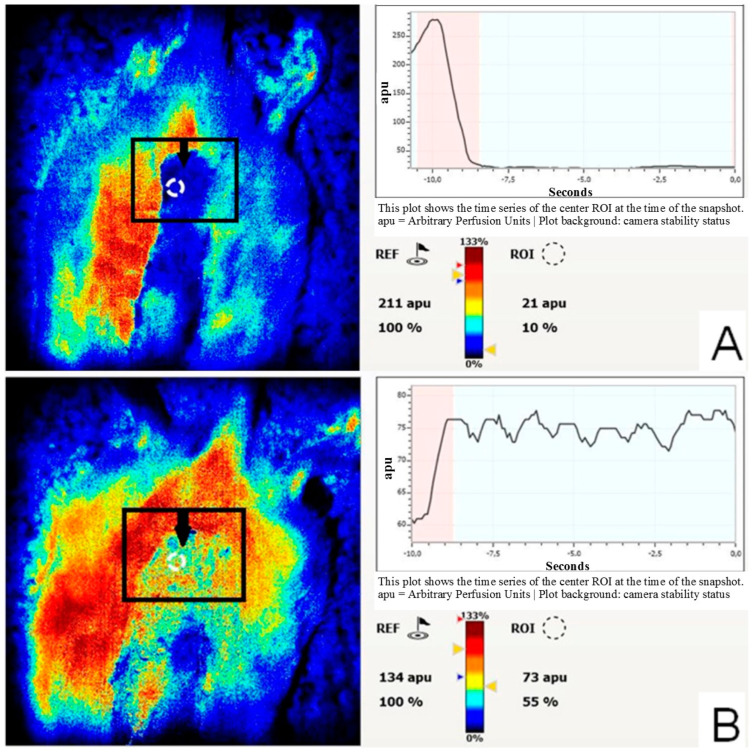
Representative intraoperative laser doppler colour images evaluated four days post-surgery. (**A**)-decompensation of the blood supply of the flap, control wounded skin without intervention; (**B**)-compensation of the blood supply of the flap, direct gene therapy wounded skin. Arrows indicate the caudal part of the flap-microcirculation assessment point. The perfusion values were reported in Arbitrary Perfusion Units (APU).

## Data Availability

The data presented in this study are available on request from the corresponding author. The data are not publicly available due to privacy.

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
