# Peer review of "Influence of Recombinant Codon-Optimized Plasmid DNA Encoding VEGF and FGF2 on Co-Induction of Angiogenesis"

_cells, 2021, doi:10.3390/cells10020432_

Round 1

Reviewer 1 Report

The paper by Salafutdinov and colleagues relies on the transplantation of hematopoietic stem cells obtained from umbilical cord blood, genetically modified with VEGF and FGF2 genes, as well as direct gene therapy with expression cassette plasmid 66 pBud-VEGF165-FGF2, during regeneration of skin wounds in rats. Authors concluded that direct application of gene therapy using VEGF and FGF2 genes stimulates rat skin wound revascularization.

This paper should be carefully revised to improve its readability

Why do authors used either 10 mm incision or 1X9 cm to evaluate the results? How they established the same criteria for comparison?

Methods could be better arranged into individual subheading to each technique.

Statistical test was not provided for the experiments. Revise it accordingly

Figure 2 is difficult to understand which is the treated group and its respective control.

Conclusion of the manuscript is lacking.

Minor comments

Conclusion of the abstract should be described in full and not numbered

It would be importante to see the mRNA expression of the cloned genes confirmed by real-time PCR. Even as supplementary material.

Ether anestesia has been abolished by the ethical commitee. Gas camera has been replaced for this purpose.

Author Response

We thank the reviewer for their careful reading of the manuscript and their constructive remarks. We have taken the comments on board to improve and clarify the manuscript. We have significantly revised and expanded the manuscript. Please find below a point-by-point response to all comments.

Reviewer 1

Comments and Suggestions for Authors

The paper by Salafutdinov and colleagues relies on the transplantation of hematopoietic stem cells obtained from umbilical cord blood, genetically modified with VEGF and FGF2 genes, as well as direct gene therapy with expression cassette plasmid 66 pBud-VEGF165-FGF2, during regeneration of skin wounds in rats. Authors concluded that direct application of gene therapy using VEGF and FGF2 genes stimulates rat skin wound revascularization.

This paper should be carefully revised to improve its readability

Why do authors used either 10 mm incision or 1X9 cm to evaluate the results? How they established the same criteria for comparison?

The study used a modified and well described reproducible model by McFarlane et al. 1965  [1097 / 00006534-196502000-00007]. We have raised this issue in the discussion. We previously tested the used model. We deem that's the presence of a small pedicle significantly complicates the process of skin flap revascularization and brings it closer to surgical practice (https://doi.org/10.1155/2018/9470198).

Methods could be better arranged into individual subheading to each technique.

We made the required changes

Statistical test was not provided for the experiments. Revise it accordingly

We made the required changes

Figure 2 is difficult to understand which is the treated group and its respective control.

We made the required changes  

Conclusion of the manuscript is lacking.

We made the required changes, conclusion added

 Minor comments

Conclusion of the abstract should be described in full and not numbered

Сomment was taken into account

It would be importante to see the mRNA expression of the cloned genes confirmed by real-time PCR. Even as supplementary material.

Сomment was taken into account, and results were added

Ether anestesia has been abolished by the ethical commitee. Gas camera has been replaced for this purpose.

We made the required changes

Reviewer 2 Report

The manoscript refers to the problem of to stimulate skin wound healing  and sets out, the use of direct gene therapy using the VEGF and FGF2 genes for stimulates the revascularization of the rat cutaneous wound.

Undoubtedly, a detailed description was made of tissue regeneration  following skin wounds in rats. The search results to have a good impact both in scientifical and clinical fields. The increased understanding of the biological characteristics, will help to clarify both the physiological mechanisms and the determination of new specific therapeutical targets in alternative to traditional therapy.

The manoscript is very ambitious despite that, it is not very well organized and is poor in discussion and this is not very good for the manoscript itself.   

Research carried out by colleagues is useful for tissue regeneration. It can be of interest in a clinical setting as it seeks to improve and speed up regenerative processes.

For this purpose, I believe it is useful to implement methods and results by associating markers of oxidative stress, inflammation, proliferation and differentiation with immunohistochemistry and Laser Doppler imaging.

Implement the conclusions in the abstract.

There is no discussion.

The conclusions alone are not exhaustive.

Author Response

We thank the reviewer for their careful reading of the manuscript and their constructive remarks. We have taken the comments on board to improve and clarify the manuscript. We have significantly revised and expanded the manuscript. Please find below a point-by-point response to all comments.

The manuscript refers to the problem of to stimulate skin wound healing  and sets out, the use of direct gene therapy using the VEGF and FGF2 genes for stimulates the revascularization of the rat cutaneous wound.

Undoubtedly, a detailed description was made of tissue regeneration  following skin wounds in rats. The search results to have a good impact both in scientifically and clinical fields. The increased understanding of the biological characteristics will help to clarify both the physiological mechanisms and the determination of new specific therapeutically targets in alternative to traditional therapy.

The manuscript is very ambitious despite that, it is not very well organized and is poor in the discussion, and this is not very good for the document itself. 

Research carried out by colleagues is useful for tissue regeneration. It can be of interest in a clinical setting as it seeks to improve and speed up regenerative processes.

For this purpose, I believe it is useful to implement methods and results by associating markers of oxidative stress, inflammation, proliferation and differentiation with immunohistochemistry and Laser Doppler imaging.

Thank you for your critical comment, we will try to do this research type in the future.  

Implement the conclusions in the abstract.

We have done made all the amendments as you requested

There is no discussion.

We made the required changes. We have added discussion.

The conclusions alone are not exhaustive.

We made the required changes. We have added discussion.

Round 2

Reviewer 1 Report

No additional comments

Reviewer 2 Report

The manuscript by Salafutdinov et al 'Influence of recombinant codon-optimized plasmid DNA encoding VEGF and FGF2 on co-induction of angiogenesis'  is very interesting and has a high value. Co-workers has considered my comments I think the authors have improved the work.  The article can be accepted.